# Impact of Partitioning in Short-Term Food Contact Applications Focused on Polymers in Support of Migration Modelling and Exposure Risk Assessment

**DOI:** 10.3390/molecules27010121

**Published:** 2021-12-26

**Authors:** Rainer Brandsch, Mark Pemberton, Dieter Schuster, Frank Welle

**Affiliations:** 1SAFE+ Algorithmics GmbH, Untere Läng 8c, 82205 Gilching, Germany; rainer.brandsch@saferithm.com (R.B.); dieter.schuster@saferithm.com (D.S.); 2Systox Limited, Sutton Grange, Parvey Lane, Sutton, Cheshire SK11 0HX, UK; markpemberton@systox.com; 3Fraunhofer Institute for Process Engineering and Packaging (IVV), Giggenhauser Straße 35, 85354 Freising, Germany

**Keywords:** food packaging, food contact material, repeated use, migration kinetics, theory and diffusion modelling

## Abstract

Food contact materials (FCMs) can transfer chemicals arising from their manufacture to food before consumption. Regulatory frameworks ensure consumer safety by prescribing methods for the assessment of FCMs that rely on migration testing either into real-life foods or food simulants. Standard migration testing conditions for single-use FCMs are justifiably conservative, employing recognized worst-case contact times and temperatures. For repeated-use FCMs, the third of three consecutive tests using worst-case conditions is taken as a surrogate of the much shorter contact period that often occurs over the service life of these items. Food contact regulations allow for the use of migration modelling for the chemicals in the FCM and for the partitioning that occurs between the FCM and food/simulant during prolonged contact, under which steady-state conditions are favored. This study demonstrates that the steady-state is rarely reached under repeated-use conditions and that partitioning plays a minor role that results in migration essentially being diffusion controlled. Domains of use have been identified within which partitioning does not play a significant role, allowing modelling based upon diffusion parameters to be used. These findings have the potential to advance the modelling of migration from repeated-use articles for the benefit of regulatory guidance and compliance practices.

## 1. Introduction

In the last two decades, significant advances have been made in migration modelling through the consideration of the underlying physical-chemical processes that occur under single and repeated-use conditions and offers opportunities for more realistic migration and exposure assessments when compared to conventional testing protocols [1,2,3,4,5,6,7,8]. These advances include the prediction of diffusion coefficients D_P_ in the polymers that are used for food contact materials and articles [2,9,10]. On the other hand, the partition coefficient influences the migration modelling results as well. However, practical prediction models that are similar to those that are used to determine diffusion coefficients in polymers are not available [11]. Octanol/water coefficients can be used for the estimation of a partition coefficient between a polymer and a food (simulant) [12,13,14]. On the other hand, advances that have been in the understanding of diffusion modelling include an appreciation that partitioning between a food contact article and a foodstuff only plays a significant role under conditions of prolonged contact, where steady-state conditions are likely to develop. In contrast, under “short-term contact” conditions, the equilibrium between the polymer and food (simulant) will be most probably not reached within the contact time. Therefore, the contribution of the partition coefficient K_P,F_ to the migration process will only play minor role. Because of the lack of real-life data and simple estimation procedures for partition coefficients in the scientific literature, such advances in diffusion modelling are enabling the industry to develop more realistic safety assessments and to define domains of safe use for repeated-use food contact polymers. The adaptation of existing regulations to recognize these developments will lead to a better understanding of the true health risks that could be associated with these materials and to the creation of opportunities that focus compliance and enforcement activities.

The application of migration modelling for compliance assessment and evaluation has been discussed by the scientific community for more than two decades and has become part of the food contact materials regulations with Commission Directive 2002/72/EC [15] in 2002. As a major outcome of this discussion, it is generally accepted that the migration of substances from (packaging) polymers follows Fick’s 2nd law of diffusion. The solution of this diffusion law provides the possibility to predict the migration of the individual substances that are present in the food contact polymers in support of its safety evaluation for single- as well as repeated-use applications, provided that the related mass transfer parameters, the diffusion coefficient D_P_, and the partition coefficient between the polymer and the food K_P/F_ are accessible [16].

Current migration modelling practices are very conservative. Diffusion coefficients D_P_ are predicted in a worst-case scenario, meaning that the predicted migration is over-estimative of real migration into food in most cases. In addition, highly over-estimating estimation procedures are applied for the partition coefficients K_P/F_. According to the European Commission Joint Research Center (JRC) Technical Guideline on Migration Modelling [13], only two values are to be used, i.e., K_P,F_ = 1 for migrants that are soluble in the food or simulant, and K_P,F_ = 1000 for migrants that are insoluble in the food or in the simulant. In view of real-life complexity, where several hundreds of different polymers and several thousands of substances are used in food contact applications, it is of major interest to determine the use scenarios in which the impact of partitioning on migration is negligible. For example, if the equilibrium between the migrant and food is at the end of the food’s shelf life, then the partitioning coefficient does not play a significant role. Due to the short contact times, this is especially the case for repeated-use articles. However, due to the authors’ knowledge of the criteria that are implemented when the partitioning, the coefficients that should be taken into account are not published in the scientific literature.

Migration modelling is more challenging for repeated-use applications than it is for single-use applications. As mentioned above, due to the lack of knowledge of the real migration behavior of potential migrants in repeated-use articles or in food packaging materials in general, highly over-estimated modelling parameters are applied. Regarding repeated-use articles, each cycle will exaggerate any differences between real parameters and any calculated parameters that are used, leading to the over-estimation of migration at early time points, and consequently, to under-estimation at later time points [8]. In the case of repeated-use articles, it is therefore necessary to apply modelling parameters that accurately describe migration into food. Such realistic modelling parameters, however, are not available for most of the packaging polymers. These realistic diffusion coefficients cannot be derived from migration kinetics into food simulants because of the slight extractive behavior of the food simulants. Therefore, testing repeated-use articles with food simulants such as 50% ethanol or 95% ethanol are often not comparable to repeated-use applications with real food because exaggerated testing conditions with such simulation solvents can lead to swelling, providing unrepresentatively high migration results compared to those that are seen during real migration into food [17,18,19]. Therefore, it is even more important to develop reliable modelling tools to address the safety evaluation of repeated-use polymers and single-use polymers with short-term contact. A comparison between experimental migration work and modelled migration showed that for defined application domains that are generally applicable to migration, including single-use applications, but that were particularly relevant to repeated-use applications with short contact times and moderate use temperatures, that partitioning has no significant impact on the migration process [17,20]. It is thus possible to predict realistic migration levels within certain domains based only on realistic diffusion coefficients D_P_ without considering the partition coefficient K_P/F_. Since it only requires diffusion coefficients D_P_ rather than partition coefficients as well, K_P,F_ makes migration modelling more accessible for industrial practical applications. However, it is still necessary to use accurate diffusion parameters for a given polymer, and this may require some investment if these are not available.

The aim of this work is to identify the conditions of use under which no equilibrium can develop and in which partitioning can therefore only have a negligible influence on the migration process for single use as well as for repeated-use articles. In these cases, the partition coefficient K_P,F_ does not need to be considered in migration modelling, and a reference partition coefficient of K_P,F_ = 1 can be used for all calculations as a worst-case scenario. This will simplify the assessment of food contact materials and the application of migration modelling, and it also provides the possibility of performing a migration evaluation or at least a migration risk assessment. This paper addresses both single-use and repeated-use cases where the diffusion process is the controlling factor for migration and looks at the principles and provides some case studies illustrating the application of this approach.

## 2. Results

### 2.1. Generic Diffusion Coefficient Approach

Within this chapter, the influence of the partition coefficient on migration was investigated using a generic approach that was based on different diffusion and partition coefficients. This approach is based on the following well-known observations:The migration rate of a substance in a polymer itself mainly depends on the diffusion rates of the migrant in the material. The diffusion rates depend on the material type (mainly mobility of the polymer at molecular level), migrant (mainly size of the migrant), and temperature (random movement induced by temperature). The diffusion rate is therefore represented by a diffusion coefficient. The diffusion coefficient can be considered to be a “material” constant for the combination of the polymer, substance, and temperature.The migrating fraction is related to the initial amount of a substance from a polymer to a contact medium and mainly depends on the partition coefficient of the migrant between the polymer and contact medium, e.g., food, simulating solvent, cosmetic or pharmaceutical formulation, drinking water, etc. The partitioning might be also temperature dependent. The partition coefficient is the constant for a polymer, substance, contact medium, and temperature.The migrating amount of the substance depends in addition on its initial concentration c_P,0_ in the polymer and the thickness of the material, i.e., the absolute amount in the system.

Based on the above-mentioned well-known observations, the migration process in this generic approach can be displayed using diffusion coefficients, partition coefficients, material thicknesses, and concentrations. Due to the fact that the migration is directly proportional to the concentration in the material, the generic approach can be simplified if the migration is displayed as a percentage of the maximum migration at equilibrium. Based on the above summarized dependencies, the migrating amount in mg/L for a single use item is plotted against the storage time between the polymer and the contact medium for different partition coefficients. For each graph, the diffusion coefficient of the migrant in the material, the initial concentration of the migrant in the material, and the material thickness have fixed values. The migration was calculated for several packaging material thicknesses (50 µm, 100 µm, 200 µm, 500 µm, and 1000 µm) with an initial concentration of c_P,0_ = 100 mg/kg in the material. The surface volume ratio was set to V = 1000 mL and A = 600 cm^2^, which represents a generic packaging, the so-called “EU-cube”. Figure 1 (thickness 50 µm) and Figure 2 (thickness 500 µm) show the predicted migration in terms of the dependence of the diffusion coefficients D_P_, which range from 10^−10^ cm^2^/s to 10^−15^ cm^2^/s, and partition coefficients, which range from K_P,F_ = 0.1 to 10,000. These parameter ranges represent typical real scenarios for food contact materials. For parameter values that are outside of this range, the impact of the partition coefficient could be different. Similar graphs for other packaging thicknesses are given in the Appendix A.

The storage time in this generic approach was assumed to be up to 365 d. Even if the topic of this publication concerns short-term contact, the migration results for storage times of up to 365 d seemed to be useful for understanding the diffusion and partitioning process in food contact materials. In case of short-term contact, the maximum storage time was below 10 d, and in most cases, it was below 1 d. 

As shown in the figures, high diffusion rates in combination with long contact times results in the partition coefficient demonstrating a strong influence on the migration results. In this case, maximum migration is reached within the storage time. As expected, the partition coefficient also has an influence on the migration amount. High partition coefficients indicate low solubility in the contact medium and therefore lower concentrations of the low partition coefficients (e.g., *K* = 1, good solubility). 

Figure 1 shows the generic approach for a thin food contact item with a material thickness of 50 µm. For graphs (a), (b), and (c), which depict high diffusion rates (D_P_ = 10^−10^ cm^2^/s to 10^−12^ cm^2^/s), the equilibrium is reached after storage for 365 d, and therefore, the partition coefficient plays the main role in the migration process. In terms of short-term contact, at the very fast diffusion coefficient of 10^−10^ cm^2^/s, the equilibrium is also reached after a storage time of 10 d. For the material thickness of 100 µm (see Appendix A), equilibrium is reached in graphs (a) and (b) after 365 d storage. All of the other diffusion coefficients result in non-equilibrium situations after 365 d of storage. After short-term storage (10 d), equilibrium is only reached for a diffusion coefficient of 10^−10^ cm^2^/s. The diffusion coefficient of 10^−11^ cm^2^/s resulted in a non-equilibrium situation for the partition coefficients of *K* = 0.1 to *K* = 100, whereas the equilibrium is still (nearly) reached for *K* = 1000 and *K* = 10,000. For a material thickness of 200 µm, the equilibrium is (nearly) reached only for *K* = 1000 and *K* = 10,000 for D_P_ = 10^−10^ cm^2^/s (see Appendix A). Similar situations were achieved with material thicknesses of 500 µm (Figure 2) and 1000 µm (see Appendix A). 

As a result, the partitioning coefficients play a main role in combination with high diffusion coefficients and thin food contact materials. When considering lower diffusion coefficients and higher material thicknesses, the diffusion coefficient plays the most important role, especially at short storage times.

In order to translate the generic graphs into a practical tool for migration prediction, the diffusion coefficients must be assigned to polymers, substances, and temperatures. For the typical polymers and migrants that are used in food contact applications at ambient temperatures, the diffusion coefficient ranges between 10^−10^ cm^2^/s (high diffusivity polymer, small molecule) and 10^−15^ cm^2^/s (low diffusivity polymer, huge molecule). Figure 3 shows examples for polypropylene (PP) [21], general-purpose polystyrene (GPPS), high-impact polystyrene (HIPS) [22], acrylonitrile butadiene styrene copolymer (ABS) [21], styrene acrylonitrile copolymer (SAN [21]), and polyethylene terephthalate (PET) [10].

The impact of the partition coefficients can be summarized using a so-called migration map (Figure 4). The migration map is generated by calculating the migration in a diffusion coefficient range from 10^−10^ cm^2^/s to 10^−16^ cm^2^/s and in a time range between 0 and 10 days (single use) independently of a partition coefficient range from *K* = 0.1 to *K* = 10,000. The migration map visualizes the influence of the partition coefficient on the migration results. As a quantitative criterion, we suggest considering the partition coefficients in migration modelling when the migration values are above 10% of the migration at equilibrium. Such a 10% criterion is supported by the fact that analytical approaches are also typically associated with a measuring uncertainty in the range of 10%. Therefore, this percentage seems to be a suitable indicator as to whether the dependence of the migration from partitioning is significant or not. In the case the migration variation being dependent on the partition coefficient being less than 10%, it is considered that the impact of partitioning on migration is of low significance. A migration variation in this range would not be accessible by experimentation due to the uncertainty usually being in the range of 10% during migration measurements. If the migration range resulting from different *K* values has a maximum migration value that is no more 10% higher than the minimum migration value, then the point is colored in light green. For single-use applications, the smallest partition coefficient *K* = 0.1 provides the highest migration result, which is therefore the basis for the migration map. If the influence is higher than 10%, the color changes to dark green at 25%, bluish green at 50%, violet at 75%, blue at 100%, and white at >100%. 

The map shows the deviation d according to Equation (1), where *m*(*K*,*t*) is the migration mass for a given *K* at the time *t*.
(1)$$d\left(t\right)=max_{K}\left(m\left(K,t\right)\right)/min_{K}\left(m\left(K,t\right)\right),$$

The migration map provides the possibility of deciding if partitioning needs to be considered at a given diffusion coefficient of the migrant in the material and at a different food contact time. At this point, it should be noted that the main factors of influence for the diffusion coefficient are the polymer type, the migrant, and the temperature. For a single-use food contact applications (Figure 4) at a diffusion coefficient of 10^−^^15^ cm^2^/s and a contact time of 2 days, the impact of partitioning on migration is of low significance (neglectable < 10%); however, for the same contact time but higher diffusion coefficient of 10^−13^ cm^2^/s, partitioning starts to become significant (75%).

Compared to single-use food contact articles, where the food is in continuous contact with the material during a food’s shelf life, in repeated-use food contact applications, the food is periodically removed from the material and is brought in contact with fresh portions of food during the service time of the material or article in question. A migration map for a repeated-use food contact application is given in Figure 5. As a result, a diffusion coefficient of 10^−18^ cm^2^/s (position 1 in Figure 5) and a contact time of 6 days (=2 × 3 days), partitioning demonstrates a neglectable impact on migration (<10%). However, for the same contact time but higher diffusion coefficient of 10^−13^ cm^2^/s (position 2 in Figure 5), partitioning plays a significant role (>100%). As seen in Figure 5, the impact of partitioning on migration is equal to that seen in the single-use graph for the first contact period (Figure 4) but differs significantly at the beginning of the second and third and following contact periods when fresh food or when a food simulant is brought in contact with the material. This behaviour can be explained by the time-dependent migration curves in Figure 6, which show time-dependent migration at a fixed diffusion coefficient (10^−^^13^ cm^2^/s) for all partition coefficients (*K* = 0.1 to *K* = 10,000). It becomes visible that immediately after bringing the material or article in contact with fresh food again, the impact of partitioning is high and lowers over time until the next repeated use occurs. In the initial phase, the migrating amount is much higher for a high partition coefficient (*K* = 10,000) compared to a low one (*K* = 0.1). The reason for this observation is that in the diffusion-controlled part of the migration curve, the migration is proportional to the migrant’s concentration. When the partition coefficient is high, the concentration of the migrant at the interface is high at the end of a contact period, and after the food has been exchanged and is assumed to be free of migrant, proportionally high migration to the food is observed. When the partition coefficient is low, the concentration of the migrant at the interface is low at the end of a contact period, and after the exchange of the food, a proportionally low migration to the food is observed. 

In summary, the impact of partitioning is high immediately after the food has been exchnaged and lowers over time until the next repeated-use occurs. In detail, after 8 days (= 192 h = 2 × 3 days + 2 days) at a diffusion coefficient of 10^−^^13^ cm^2^/s (position 3 in Figure 5), the two migration curves for *K* = 0.1 and *K* = 10,000 cross each other and stay close to each other throughout the remaining time of the contact period. Immediately after the exchange of the food, the migration for a high partition coefficient is higher compared to migration at a low partition coefficient, becomes equal, and exceeds it at a later time point.

### 2.2. Material, Temperature and Migrant Specific Approach

The main deficiency of the generic “short-term contact” considerations that are solely based on diffusion coefficients and partition coefficients if relevant is the missing link to real case applications where migration assessment is material, substance, and temperature dependent. This limitation of the generic approach can be exemplified by the following two case examples for PET computed with the Welle estimation procedures [10]: (i) A molecule with a molecular volume of M_V_ = 195 Å^3^ at a temperature of T = 40 °C results in (nearly) the same diffusion coefficient of D = 7.8 × 10^−17^ cm^2^/s as a molecule with M_V_ = 326 Å^3^ at T = 60 °C. (ii) Substances with M_V_ = 130 Å^3^ at T = 70 °C and M_V_ = 65 Å^3^ at T = 30 °C both have a diffusion coefficient of about D = 2.3 × 10^−13^ cm^2^/s. Decomposing the diffusion coefficient into the contributions of the material, substance, and temperature makes a graphical representation very complex and impractical. Therefore, in the following section, an attempt is made to show the dependence of amount of migration from a substance resulting from a partition coefficient between the polymer and food for different time points at a fixed temperature and molecular mass (equivalent to size of migrant). These results are given in Figure 7. The significant dependence of the migration from the partition coefficient becomes visible in Figure 7 by drop of the curve, i.e., a decrease in migration as the partition coefficient increases. For a small molecule (50 g/mol), the impact of partitioning on migration becomes relevant at the partition coefficient *K* = 1000, and for bigger molecules (250 g/mol), the impact of partitioning in impact becomes relevant at the partition coefficient *K* = 100,000. A benefit of this representation is that the migrating amount is included in the graph, which means that a compliance evaluation with a restriction or a risk assessment that is based on toxicological considerations are possible.

In the following figures, the migration maps for various polymers (PET, ABS, PA6 and PP) are presented and discussed. These maps were generated from Figure 7 using the diffusion coefficients available from Figure 3. Figure 8 show the deviation in the migration in mg/dm² that is independent of the partition coefficient (K_P,F_) that is expressed in % at a given initial concentration (1000 mg/kg), material thickness (2000 µm), and area to volume ratio (6 dm^2^/kg) for PET. For PET and the contact times under 10 days, the impact of partitioning on migration is neglectable (<10%) at the temperature of 40 °C and with a migrant with a molecular mass of 200 g/mol, which is significant at 100 °C. For a comparison of a single-use scenario (10 days) with a repeated-use scenario, the migration maps were generated for the time period of 6-9 days from three consecutive uses of 3 × 3 = 9 days. The main difference is that under repeated-use conditions, the partition coefficient is significant for the whole range of migrants and temperatures investigated, i.e., the migration range is higher than 10% compared to the migration value for the lowest partition coefficient.

The migration map for ABS is generated by calculating the migration for a fixed polymer and for contact times of up to 10 days for a given application temperature and molecular mass of the migrant. This map is shown in Figure 9. For ABS and contact times less than 10 days, the impact of partitioning on migration is negligible (<10%) at the temperature of 60 °C and with a migrant with a molecular mass of 100 g/mol, which becomes significant at temperatures above 80 °C.

The migration map for PA6 is given in Figure 10. For PA6 and contact times of less than 10 days, the impact of partitioning on migration is significant (>10%) for the whole temperature range from 0 to 80°C and for the molecular mass range of 50 to 500 g/mol. For contact times of less than 1 day, the impact of partitioning on migration is not significant for bigger molecules (>300 g/mol) and for refrigerated applications.

The migration map for PP is given in Figure 11. For PP and contact times of less than 1 day, the impact of partitioning on migration is highly significant (>75%) for the whole temperature range of 0 to 20 °C and for the molecular mass range of 400 to 500 g/mol. For contact times below 6 h, the impact of partitioning on migration is less significant for bigger molecules (>400 g/mol) and for refrigerated applications.

## 3. Discussion

There is a need to maintain scientific approaches to evaluate the safety of polymers that are intended to be in contact with food beyond historically established experimental testing, which is time consuming and expensive. On the other hand, it is indispensable to maintain regulation approaches that are in line with recent scientific developments. Current regulation approaches are necessarily conservative but are not always reflective of real-life situations. This is the specific case for repeated-use food contact applications, in which contact time is predominantly short and service life is predominantly long compared to single-use applications.

Using modern science to evaluate the safety of repeated-use food contact polymers will provide opportunities to better evaluate and enforce the safety of polymers, and this will help to support the validation of compliance and build confidence with stakeholders. A lot of developments have taken place that improve our understanding of how to better predict the safety of polymers. The solution to the diffusion law provides the possibility to predict the migration of individual substances that are present in the food that is contact with plastic, allowing polymers to be evaluated as being safe to use for single- and repeated-use applications, provided that the related mass transfer parameters for diffusion and the partition coefficient are accessible. Meanwhile, several generally applicable estimation procedures to determine the diffusion coefficients of the migrants in polymers exist were validated by a significant number of experimental values [9,10,13], but no generally applicable and simple estimation procedure to determine the partition coefficients between the plastic food contact layer and food are available. Under these circumstances, it is of particular interest to determine under which circumstances knowledge of the related partition coefficient is of minor importance.

This study determined that the impact of the partition coefficient between the plastic food contact layer and food on migration from the plastic to food might be negligible under certain conditions, i.e., the impact of the partition coefficient on the migrating amount is less than 10%. This is the case for low diffusivity polymers such as styrene-based polymers (PS, ABS, SAN, etc.) or polyesters (PET, etc.) under short-term applications (<1 day) at temperatures below of 40 °C or those that are generically assumed for polymers with a glass transition temperature exceeding 70 °C. On the other hand, for high diffusivity polymers, i.e., polymers with a glass transition temperature below room temperature, such as polyolefins, the influence of the partition coefficient is negligible for contact times of less than 1 h at or below 20 °C. Knowing the boundaries under which the impact of the partition coefficient on the migrating amount is negligible simplifies and hence promotes the use of migration modelling for the compliance evaluation of polymers that are in contact with food. 

If the significance of the partition coefficient is low, then neglecting the partition coefficient in the simulation leads to an over-estimation of the migration amount due to an over-estimated diffusion coefficient. In case the partition coefficient is neglected, i.e., is considered to be low (good solubility of the migrant in the food or simulant), despite the fact that its impact on migration is significant, overestimated migration results and hence the legal requirement to never underestimate migration by modelling are fulfilled.

### 3.1. Impact of Repeated-Use Scenarios 

The current approach that is appropriate for single use food contact materials or article situations is not appropriate for repeated-use articles and needs to be re-evaluated. Compliance evaluations that are based on the third consecutive migration period might not be appropriate due to less frequent, i.e., intermittent use repeated-use food contact articles such as kitchen utensils or continuous repeated-use food processing equipment that is used more than three times.

The experimental work published by Mercea et al. [8] has been used as example and as point of departure for investigations into the relevance of intermittent use scenarios is given in Figure 12 for Metilox (3-(3,5-Di-tert-butyl-4-hydroxyphenyl)propionic acid methylester, CAS 6386-38-5) and Irganox 1076 (3-(3,5-Di-tert-butyl-4-hydroxyphenyl)propionic acid octylester, CAS 2082-79-3). The orange line in Figure 12 reproduces experimental results for daily use (=intermittent use), the blue line represents no empty storage (=continuous use), and the green line represents a use once per week (=occasional use). Using this published work on intermittent use as an example, it can be shown that compared to repeated-use and occasional use, i.e., varying the time without food contact (empty time), significant differences in migration can be expected. The longer the empty time is, the closer the repeated-use migration comes to the single-use migration. It is anticipated that a systematic investigation into the impact of the empty migration time would be meaningful, with potential amendments to the legislation that is currently in place being one of its benefits. The cumulated migration results without “no-contact” times are given in Figure 13. 

From the repeated-use migration scenarios simulated in Figure 12 and Figure 13, it can be concluded that continuous uses that occur one after the other, such as for a piece of food processing equipment, lead to a continuous decrease in the migrating amount. The decrease is faster if the migrant exhibits good solubility in the food (low partition coefficients) and lower in the case of a low-solubility migrant in the food. In case of intermittent use (e.g., 8 h contact of the material or article with the food followed by 16 h without food contact), the decrease in the migrating amount over time is lower compared to that of continuous use, and for materials or articles that are occasionally over time, then the migration amount stays close to the migration amount seen for a single-use application.

### 3.2. Impact of Swelling

Experimental compliance testing foresees use of simulants that are inappropriate in many cases and that the polymer swell in a way that is not representative of normal use [23]. This is especially the case for food simulants of similar polarities as the polymers that are tested and becomes of increasing relevance with increasing temperatures. An example of a practice that tends to suffer from polymer-swelling effects is testing styrene-based polymers with ethanol (alternative food simulant for vegetable oil) or ethanol–water mixtures (50% ethanol in water as food simulant for dairy products) [19]. 

In practice, materials will come into contact with foods, which may cause the polymer to swell due to the polymer’s uptake of food components. An example of this would be how non-polar polymers such as polyolefins take up vegetable oil, how polar polymers such as polyamides take up water, or how medium-polar polymers such as polyesters take up alcohol. Swelling involves plasticizing the polymer by lowering its glass transition temperature, such as by intentionally adding plasticisers to polymers, e.g., adding 15%, 30%, or 50% diethylhexylphthalate (DEHP) to PVC or PVC/PMMA blends [24]. A systematic study on the effect of the swelling of official food simulants on ABS was published by Guazzotti et al. [19]. From their data, it was concluded that the use of high concentrations of ethanol–water, especially at high temperatures, causes ABS to swell and results in significantly higher migration values compared to real foods. Similar swelling effects can be observed for PET when testing with ethanol 95% or ethanol 50% [25]. Brandsch published a linear correlation of the polymer-specific constants of polymers with glass transition temperatures [13]. According to Begley et al. [9], the polymer-specific constant is the main factor that determines the pre-exponential factor in the estimation procedure based on the Arrhenius relationship for the diffusion coefficients of the migrants in polymers. In summary, polymer swelling will lower its glass transition temperature and will consequently increase the diffusion rates of the migrants that are in the swollen polymer.

According to Wypych [26], the glass transition temperature (Tg) of polyamide 6 strongly depends on humidity, i.e., the experimental values of the glass transition temperature for dry polyamide 6 are in the range from 50 to 75 °C; if they are exposed to 50% relative humidity, then Tg-values are in the range from 3 to 20 °C; and if they are exposed to 100% relative humidity, then Tg-values are in the range of −22 to −32 °C. The diffusion coefficients of the migrants in PA6 can be estimated independently of the degree of swelling that is related to relative humidity based on the linear correlation of the polymer-specific constants of the polymers with their glass transition temperatures. [13]. The results of an estimation that considered the impact of humidity on migration from PA6 are given in Figure 14. A further phenomenon that is already known qualitatively is the impact of material swelling by the regulated simulants on migration. According to today’s knowledge, material swelling that is caused by simulants increases migration rates. The impact of partitioning on migration will become significant at lower temperatures and/or smaller molecules, as shown for PA6 when it is in contact with water. The swelling effect of simulants on materials is not completely understood at the quantitative level yet, and hence further scientific work is anticipated.

## 4. Materials and Methods

### 4.1. Migration Model

Migration was modelled based on Fick’s 2nd law of diffusion for a plane sheet in contact with a medium under single-use and repeated-use scenarios. The migration simulation was performed using the AKTS SML software version 4.54 (AKTS AG, Siders, Switzerland) and the migraSIM modelling web service (SAFE+ Algorithmics GmbH, Gilching, Germany). AKTS SML was developed by Roduit et al. [27]. migraSIM is a new solver and was developed by Brandsch and Schuster [28]. The AKTS SML software is based on finite differences methods, whereas the migraSIM service uses finite volume methods. For the simulations that were performed in this study, a generic food contact material with 1000 mL of food in contact with 600 cm^2^ of material was assumed. A range of material thicknesses at a fixed initial migrant concentration was considered for the migration calculations. As a generic approach, instead of specific substances, the materials and conditions of the use diffusion coefficients were considered. Diffusion coefficients can be translated into practical cases if the substance, the polymer, and the temperature are specified.

### 4.2. Numerical Methods

The numerical solver migraSIM can handle multiple layers of polymers and food simulants with different diffusion coefficients by solving Fick’s 2nd law of diffusion numerically, according to Equation (2).
(2)$$\frac{dc}{dt}=\frac{-\partial }{\partial x}D_{j}\frac{\partial c}{\partial x}$$

In Equation (2), *c* is the concentration of a substance (or migrant) in the matrix. A short description of the two-layer case is provided, i.e., *j* is 1 or 2, as this article is restricted to two layers: one for the polymer and one for the food. The matrix of the polymer and food layer can be restricted to a one-dimensional problem that is orthogonal to the different layers because the mass of migration is homogenously distributed along each layer. Let l be the thickness of all of the layers (Ω), and in the domain Ω = [0,l], the layer is divided into n control volumes V_1,…,V_n, where V_1,…,V_*k* belong to the polymer layer and where V_(*k*+1),…,V_n belongs to the food layer. Partitioning between the polymer and the food is modeled with a partitioning coefficient *K* that has been defined for the side condition (Equation (3)).
(3)$$\frac{\underset{}{lim}c_{k}\left(x_{b}\right)}{\underset{}{lim}c_{k+1}\left(x_{b}\right)}=K$$

In Equation (3), *x_b_* is the position of the boundary between V *k* and V (*k*+1). The discretization of Equations (2) and (3) uses a 2nd order Finite Volume scheme in space and a 1st order scheme in time. The resulting coupled system of the linear equations is solved fully implicitly, which makes larger time steps possible because of the sparse lower-upper decomposition, which is an optimized variant of the Gaussian elimination method [29]. The thinner control volumes are chosen near the polymer/food boundary in order to increase accuracy.

### 4.3. Estimation of Diffusion Coefficients

To estimate the diffusion coefficients D_P_ of the migrants in the polymers, the Piringer approach [9] as well as the Welle approach [10] were used, respectively. Both estimation procedures are based on the Arrhenius relationship. The main difference between the two estimation procedures is the use of molecular volumes by Welle compared to the use of molecular weights by Piringer. In addition, the Welle approach uses substance-specific activation energies compared to a reference activation energy used by Piringer. According to Welle, molecular volumes can be converted into molecular weights by multiplication with a factor of 1.13 [10]. Consequently, the diffusion coefficients that are estimated with the two approaches may differ, but the overall conclusions of the publication are not altered.

## 5. Conclusions

Within this study, the role of the partition coefficient in diffusion modelling was systematically evaluated for food contact articles that are in short-term contact with food. From the results, a criterion was derived for which the partition coefficient only plays a minor role in the diffusion process has a consequent effect on the results of the diffusion modelling process. Minor role means that the impact of a wide range of partition coefficients on the migration modelling results is less than 10%. This 10% value can be also considered as the typical uncertainty value in experimental migration testing, below which an impact of partitioning on migration would not be accessible in practice. 

From the generic approach used and from the correlations that were presented in this study, the temperature and molecular weight ranges were estimated, and it was determined that the partition coefficient plays a minor role in short-term for single-use food contact applications that are less than one day in duration. In summary, the impact of the partition coefficient on migration results in the diffusion modelling results for short-term food contact applications being able to be expressed as polymer-specific temperature and as molecular weight ranges. These temperature and molecular weight ranges are summarized in Table 1. The estimated values were derived for the four investigated polymers from the light green areas in the “migration maps”. For low-diffusivity polymers with glass transition temperatures > 70 °C, such as ABS (or other styrenic polymers such as GPPS, HIPS, SAN, etc.) or PET (or other aromatic polyesters) at ambient temperature and below the impact of the partition coefficient on migration plays a minor role for almost all molecules. In contrast, for high-diffusivity polymers such as PP (or other polyolefins like LDPE, HDPE, etc.) with glass transition temperatures < 20 °C, the impact of the partition coefficient on migration is relevant for all molecules with molecular weights below 1000 g/mol. For medium-diffusivity polymers (glass transition temperatures 20 °C < Tg < 70 °C ) such as PA6 (or other polyamides or aliphatic polyesters such as PBT, etc.) the impact of partitioning on migration is relevant for small molecules with molecular weights below 400 g/mol, especially at higher temperatures. For these medium-diffusivity polymers, the partition coefficient plays a minor role for bigger molecules with molecular weights above 400 g/mol. In these cases, modeling work should be performed with different partition coefficients in order to investigate the influence of the migration calculations on the partition coefficients. If they are available, known partition coefficients should be used for migration modelling. 

Regarding repeated use, the migration maps were generated for the time period of 9 days from three consecutive uses of 3 × 3 = 9 days. The main difference compared to single-use contact is that under repeated-use conditions, the partition coefficient was significant for the whole range of migrants and temperatures investigated, i.e., the migration range is higher than 10% compared to the migration value for the lowest partition coefficient. It becomes obvious that immediately after bringing the material or article in contact with food more than once, then the impact of partitioning is high and lowers over time until the next repeated use occurs. In the initial phase, the migrating amount is much higher for a high partition coefficient (*K* = 10,000) compared to a low partition coefficient (*K* = 0.1). Consequently, for the number of repeated uses calculated according to the testing protocol for food contact materials legislation in Europe, the impact of the partition coefficient on migration is higher for repeated-use compared to single-use applications.

## Figures and Tables

**Figure 1 molecules-27-00121-f001:**
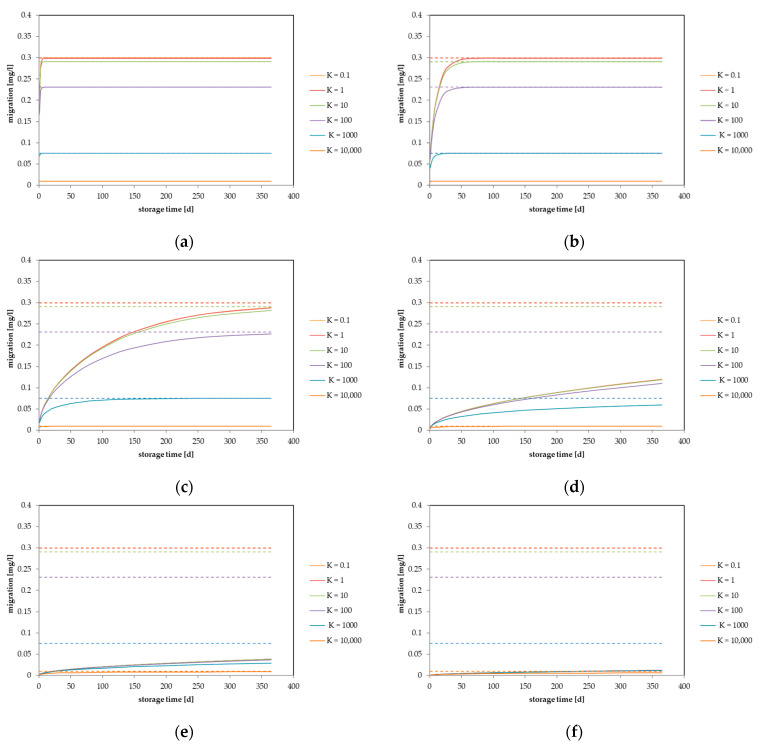
Calculated migration into food considering different partition coefficients: (**a**) diffusion coefficient D_P_ = 1 × 10^−10^ cm^2^/s, (**b**) D_P_ = 1 × 10^−11^ cm^2^/s, (**c**) D_P_ = 1 × 10^−12^ cm^2^/s, (**d**) D_P_ = 1 × 10^−13^ cm^2^/s, (**e**) D_P_ = 1 × 10^−14^ cm^2^/s and (**f**) D_P_ = 1 × 10^−15^ cm^2^/s. Calculated for a layer thickness of d = 50 µm, c_P,0_ = 100 mg/kg, V = 1000 mL, A = 600 cm^2^.

**Figure 2 molecules-27-00121-f002:**
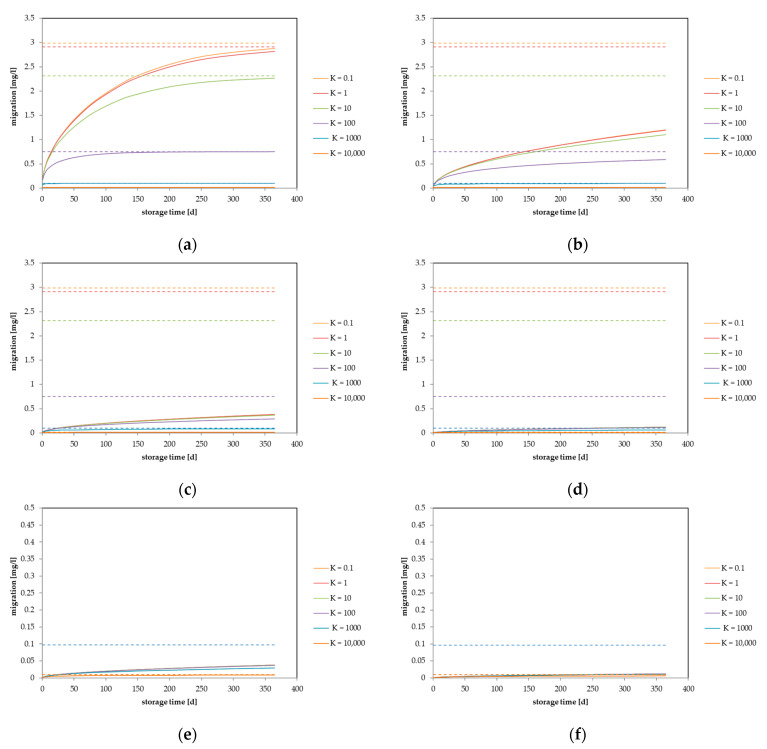
Calculated migration into food considering different partition coefficients: (**a**) diffusion coefficient D_P_ = 1 ×10^−10^ cm^2^/s, (**b**) D_P_ = 1 ×10^−11^ cm^2^/s, (**c**) D_P_ = 1 ×10^−12^ cm^2^/s, (**d**) D_P_ = 1 ×10^−13^ cm^2^/s, (**e**) D_P_ = 1 ×10^−14^ cm^2^/s and (**f**) D_P_ = 1 ×10^−15^ cm^2^/s. Calculated for a layer thickness of d = 500 µm, c_P,0_ = 100 mg/kg, V = 1000 mL, A = 600 cm^2^. Note: (**e**) and (**f**) different scale on y-axis.

**Figure 3 molecules-27-00121-f003:**
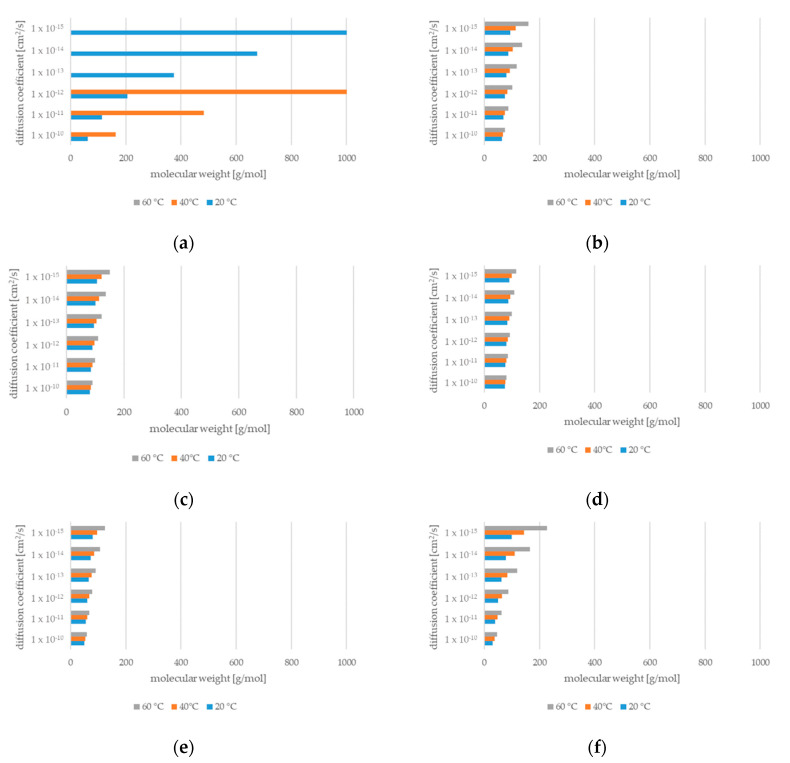
Diffusion coefficients at 20 °C, 40 °C, and 60 °C in different polymers: (**a**) PP, (**b**) GPPS, (**c**) HIPS, (**d**) ABS, (**e**) SAN, and (**f**) PET (without bars: molecular weight > 1000 mg/mol).

**Figure 4 molecules-27-00121-f004:**
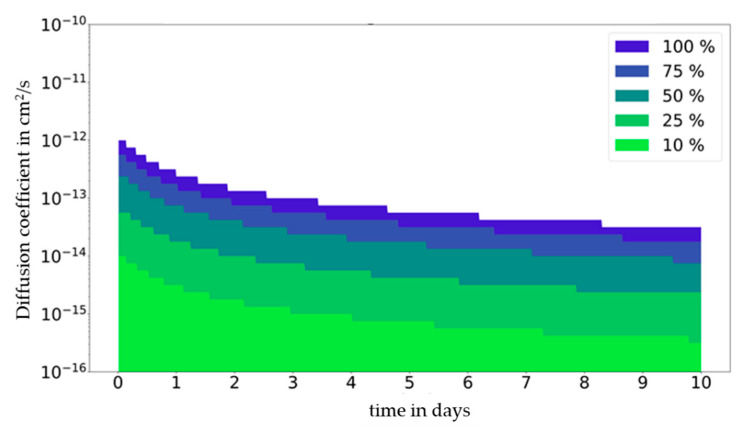
Single-use: Map of time (t) in days against diffusion coefficient (D_P_) in cm^2^/s. Deviation of migration in mg in dependence from partition coefficient (K_P,F_) expressed in % at a given initial concentration (1000 mg/kg), material thickness (2000 µm), and area to volume ratio (6 dm^2^ per kg food).

**Figure 5 molecules-27-00121-f005:**
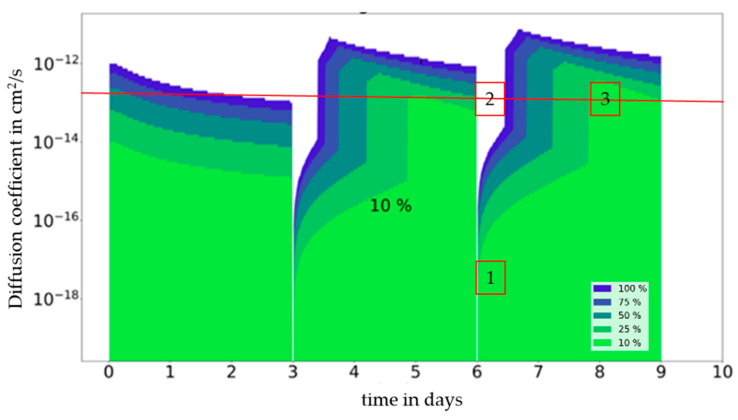
Repeated-use: Map of time (t) in minutes against diffusion coefficient (D_P_) in cm²/s. Deviation of migration in mg in dependence from partition coefficient (K_P,F_) expressed in % at a given initial concentration (1000 mg/kg), material thickness (2000 µm), and area to volume ratio (6 dm^2^ per kg food). For a description of positions 1 to 3 see text.

**Figure 6 molecules-27-00121-f006:**
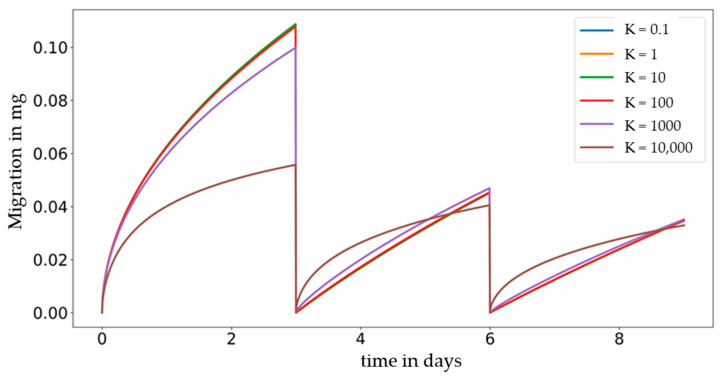
Repeated-use 3 × 72 h (= 3 × 3 days) with diffusion coefficient 10^−13^ cm^2^/s and partition coefficient *K* = 0.1 to *K* = 10,000.

**Figure 7 molecules-27-00121-f007:**
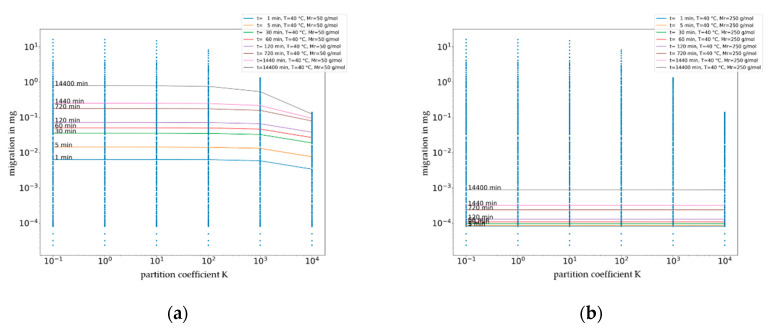
Plot of migration (m) in mg for a generic migrant of molecular weight of 50 g/mol (**a**) and 250 g/mol (**b**) from a PET layer against partition coefficient (K_P,F_) at 40 °C in dependence of time.

**Figure 8 molecules-27-00121-f008:**
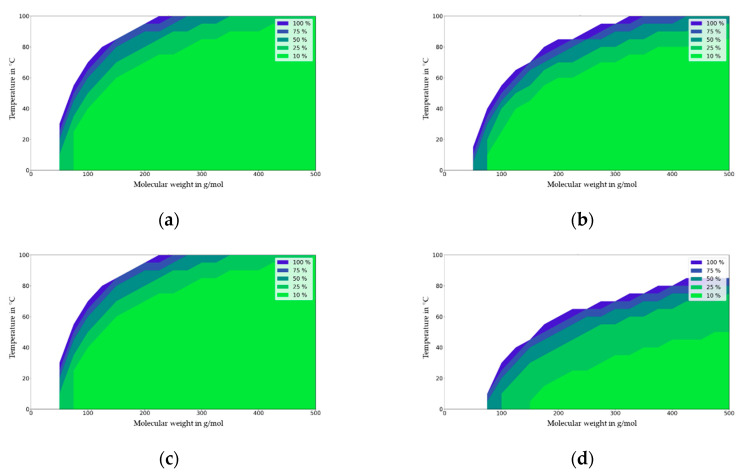
Single-use migration map for PET for contact times of 1 day (**a**) and 10 days (**b**) repeated-use migration map for contact times of 3 × 1 day (**c**) and 3 × 3 days = 9 days (**d**). Molecular weight in g/mol against temperature (T) in °C.

**Figure 9 molecules-27-00121-f009:**
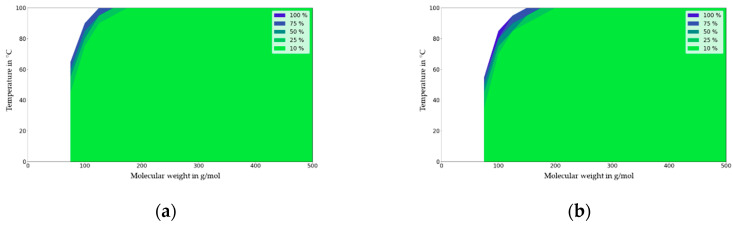
Migration map for ABS for contact times of 1 day (**a**) and 10 days (**b**). Molecular weight in g/mol against temperature (T) in °C. Deviation of migration in mg/dm^2^ independent of partition coefficient (K_P,F_) expressed in % at a given initial concentration (1000 mg/kg), material thickness (2000 µm), and area to volume ratio (6 dm^2^ per kg food).

**Figure 10 molecules-27-00121-f010:**
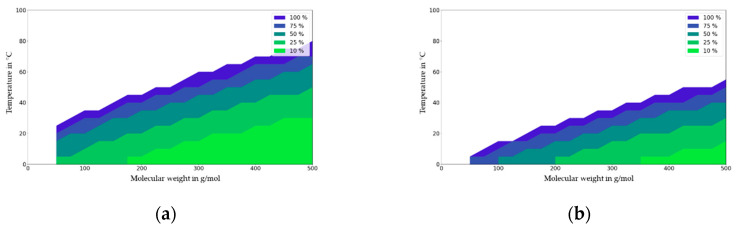
Migration map for PA6 for contact times of 1 day (**a**) and 10 days (**b**). Molecular weight in g/mol against temperature (T) in °C. Deviation of migration in mg/dm^2^ independent of partition coefficient (K_P,F_) expressed in % at a given initial concentration (1000 mg/kg), material thickness (2000 µm), and area to volume ratio (6 dm^2^ per kg food).

**Figure 11 molecules-27-00121-f011:**
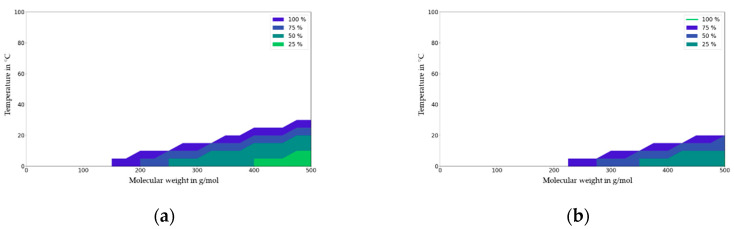
Migration map for PP for contact times of 6 h (**a**) and 1 day (**b**). Molecular weight in g/mol against temperature (T) in °C. Deviation of migration in mg/dm^2^ independent of partition coefficient (K_P,F_) expressed in % at a given initial concentration (1000 mg/kg), material thickness (2000 µm), and area to volume ration (6 dm^2^ per kg food).

**Figure 12 molecules-27-00121-f012:**
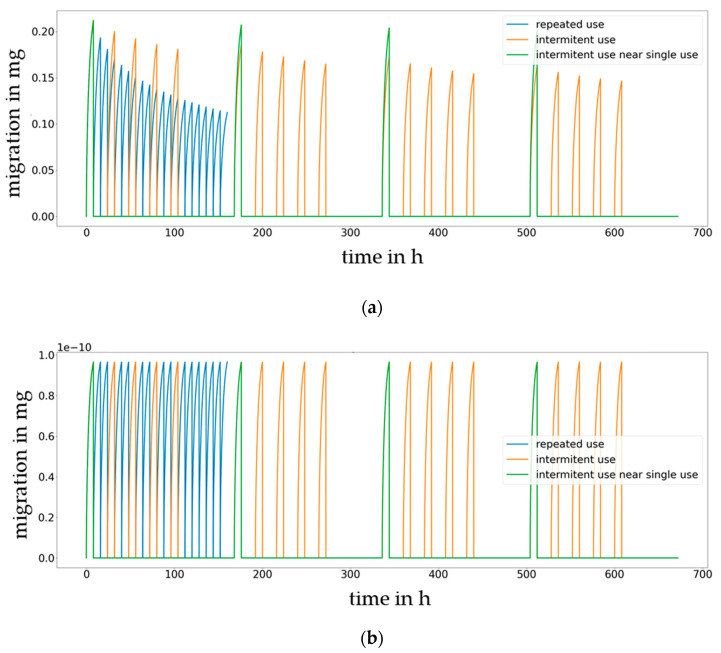
Comparison of repeated-use scenarios (blue: continuous use, orange: intermittent use 8 h contact and 16 h without contact and green: occasionally use). Migration of Metilox (**a**) and Irganox 1076 (**b**) to 50% ethanol (*K* = 10) at 60 °C.

**Figure 13 molecules-27-00121-f013:**
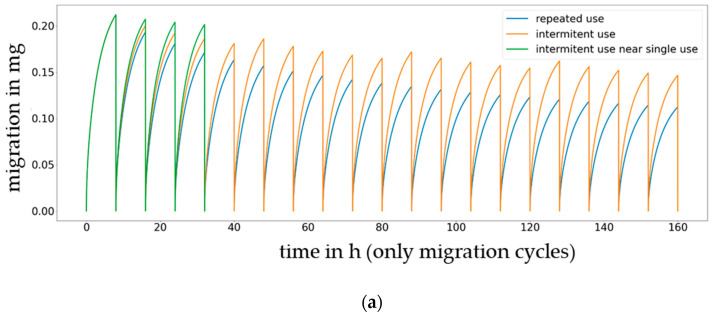
Cumulated migration of Metilox (**a**) and Irganox 1076 (**b**) to 50% ethanol at 60 °C by eliminating no-contact times (only migration cycles displayed, blue: continuous use, orange: intermittent use with 8 h contact and 16 h without contact and green: occasionally use).

**Figure 14 molecules-27-00121-f014:**
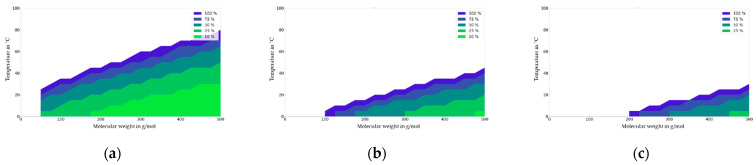
Migration map for PA6 for contact times of 1 h. (**a**) 0% relative humidity, (**b**) 50% relative humidity and (**c**) 100% relative humidity. Molecular weight in g/mol against temperature (T) in °C. Deviation of migration in mg/dm^2^ in dependence from partition coefficient (K_P,F_) expressed in % at a given initial concentration (1000 mg/kg), material thickness (2000 µm) and area to volume ration (6 dm^2^ per kg food).

**Table 1 molecules-27-00121-t001:** Estimation of temperature and molecular weight ranges in which partitioning plays a minor role in short-term single-use food contact applications.

Polymer	Tg [°C]	Refrigerated Temperatures		Ambient Temperatures		Increased Temperatures		High Temperatures	
		T (°C)	Mw (g/mol)	T (°C)	Mw (g/mol)	T (°C)	Mw (g/mol)	T (°C)	Mw (g/mol)
PET	70			<40	>100	40–70	>250	70–100	>500
ABS	100			<40	>50	40–70	>100	70–100	>200
PA6	55	<20	>400	20–40	>1000				
PP	−20	<20	>1000						

## Data Availability

The data used to support the findings of this study are included within the article.

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
