# Peer review of "Impact of Partitioning in Short-Term Food Contact Applications Focused on Polymers in Support of Migration Modelling and Exposure Risk Assessment"

_molecules, 2021, doi:10.3390/molecules27010121_

Round 1

Reviewer 1 Report

The article presents a systematic study of the migration of molecules from polymeric materials using a theoretical approach aimed at understanding when both the partition and diffusion coefficients are to be introduced in the modeling. The manuscript is well written and very clear in the introductory part. On the other hand, I found that the part of materials and methods for those who are not totally familiar with the specific field takes a lot of information for granted.

Surely all the cases investigated provide information but the choice to present so many graphs makes it difficult to follow the speech and grasp the result.

Consequently, I think that better detailing the part of materials and methods (some variables or symbols are not defined) and perhaps moving a part of the graphics into the supplementary info would make the text smoother.

I have only one specific point: - page 2 lines 69-70 I don't understand the sentence

Author Response

Reviewer 1:

The article presents a systematic study of the migration of molecules from polymeric materials using a theoretical approach aimed at understanding when both the partition and diffusion coefficients are to be introduced in the modeling. The manuscript is well written and very clear in the introductory part. On the other hand, I found that the part of materials and methods for those who are not totally familiar with the specific field takes a lot of information for granted.

Response of the authors: We went through the materials and methods part and try to make it easier for the reader.

Surely all the cases investigated provide information but the choice to present so many graphs makes it difficult to follow the speech and grasp the result.

Consequently, I think that better detailing the part of materials and methods (some variables or symbols are not defined) and perhaps moving a part of the graphics into the supplementary info would make the text smoother.

Response of the authors: Thanks for this comment. We created a Supplementary data document with parts of the graphs from the original manuscript.

I have only one specific point: - page 2 lines 69-70 I don't understand the sentence

Response of the authors: Thanks for this comment. The sentence was indeed not correct. We corrected the sentence in the manuscript.

Reviewer 2 Report

The article entitled: Impact of Partitioning on Polymer Focused Short-Term Food Contact Applications in Support of Migration Modeling and Exposure Risk Assessment presents advances in the field of knowledge and is well-written, well-modeled, and scientifically sound. I have some suggestions for improving the manuscript:
Throughout the text correct the measurement units used, ie, where is mg/kg use mg kg-1, cm2/s = cm2 s-1, g/mol = g mol-1, check the style of all units .
I suggest improving the quality of figure 8, as the numbers with the different % are on top of each other.

Author Response

Reviewer 2:

The article entitled: Impact of Partitioning on Polymer Focused Short-Term Food Contact Applications in Support of Migration Modeling and Exposure Risk Assessment presents advances in the field of knowledge and is well-written, well-modeled, and scientifically sound. I have some suggestions for improving the manuscript:

Throughout the text correct the measurement units used, ie, where is mg/kg use mg kg-1, cm2/s = cm2 s-1, g/mol = g mol-1, check the style of all units .

Response of the authors: Thanks for this comment. The use of mg/kg instead of mg kg‑1 is quite usual in the Journal. We checked a couple of manuscripts. So we decide not to change the units into the mg kg‑1 format.

I suggest improving the quality of figure 8, as the numbers with the different % are on top of each other.

Response of the authors: Thanks for this comment. We improved the Figure 8 (now Figure 5).